

# Saccades and handedness interact to affect scene memory

Timothy M. Ellmore, Bridget Mackin and Kenneth Ng

Department of Psychology, City University of New York, City College, New York, NY, United States of America

## ABSTRACT

Repetitive saccades benefit memory when executed before retrieval, with greatest effects for episodic memory in consistent-handers. Questions remain including how saccades affect scene memory, an important visual component of episodic memory. The present study tested how repetitive saccades affect working and recognition memory for novel scenes. Handedness direction (left–right) and degree (strong/consistent vs. mixed/inconsistent) was measured by raw and absolute laterality quotients respectively from an 8-question handedness inventory completed by 111 adults. Each then performed either 30 s of repetitive horizontal saccades or fixation before or after tasks of scene working memory and scene recognition. Regression with criterion variables of overall percent correct accuracy and $d$-prime sensitivity showed that when saccades were made before working memory, there was better overall accuracy as a function of increased direction but not degree of handedness. Subjects who made saccades before working memory also performed worse during subsequent recognition memory, while subjects who fixated or made saccades after the working memory task performed better. Saccades made before recognition resulted in recognition accuracy that was better (Cohen's $d = 0.3729$), but not significantly different from fixation before recognition. The results demonstrate saccades and handedness interact to affect scene memory with larger effects on encoding than recognition. Saccades before scene encoding in working memory are detrimental to short- and long-term memory, especially for those who are not consistently right-handed, while saccade execution before scene recognition does not appear to benefit recognition accuracy. The findings are discussed with respect to theories of interhemispheric interaction and control of visuospatial attention.

Corresponding author
Timothy M. Ellmore,
tellmore@ccny.cuny.edu

# INTRODUCTION

## Saccade induced retrieval enhancement

As few as thirty seconds of horizontal bilateral saccadic eye movements before testing selectively enhances explicit memory, most notably episodic memory retrieval for laboratory and everyday events (*Christman et al., 2003*). This has been termed saccade induced retrieval enhancement (SIRE) by *Lyle, Logan & Roediger 3rd (2008)*. The benefits of SIRE appear to be specific to retrieval. Early preliminary findings reported that eye movements before encoding hurt subsequent memory performance (*Christman & Butler, 2005*). Later experiments by others showed that the benefits of horizontal eye movements
are seen when they immediately precede episodic memory retrieval but not when they precede encoding (*Brunye et al., 2009*). One hypothesis is that horizontal eye movements enhance interhemispheric interaction, which is associated with superior episodic memory (*Christman & Propper, 2001*). Increased interhemispheric interaction has also been related to decreased false memories in a semantic associates paradigm (*Christman, Propper & Dion, 2004*).

Bilateral eye movements also appear to enhance some types of recognition memory. Subjects who made bilateral eye movements were more likely to correctly recognize previously presented words, but less likely to falsely recognize critical non-studied associates (*Parker & Dagnall, 2007*). *Parker, Relph & Dagnall (2008)* conducted multiple experiments in a study investigating effects of bilateral saccadic eye movements on item, associative, and contextual information. In tests of item recognition they found bilateral horizontal eye movements versus no eye movements enhanced item recognition by increasing hit rate and decreasing false alarms; an additional remember-know analysis showed eye movements increased "remember" responses. For associative recognition, they found bilateral eye movements increased correct responses to intact pairs and decreased false alarms to rearranged pairs (*Parker, Relph & Dagnall, 2008*). Bilateral eye movements also increased correct recall for both intrinsic (color) and extrinsic (spatial location) context (*Parker, Relph & Dagnall, 2008*).

More recently, the horizontal SIRE effect has been extended beyond the visual domain in a study that showed retrieval enhancement in the somatosensory system after alternating left–right tactile stimulation (*Nieuwenhuis et al., 2013*). In what appears to be the only study of the effects of horizontal saccadic eye movements on the retrieval of landmark shape and location information, increased recognition sensitivity and decreased response times were reported in a spatial memory test with effects only seen when eye movements preceded episodic memory retrieval, but not when they preceded encoding (*Brunye et al., 2009*). Furthermore, these same authors found that eye movements were only beneficial in an old-new recognition paradigm, which purportedly involves more elaborate recollective processing and presumably demands a high degree of right and left-hemisphere activity, compared to a forced-choice recognition test, which is thought to be more dependent on familiarity.

Despite the significant clinical and applied implications for the effects of bilateral saccades (*Lyle & Jacobs, 2010*; *Propper & Christman, 2008*; *Stickgold, 2002*), there are only a few studies employing neuroscientific methods to measure brain-based correlates of bilateral saccades. Bilateral eye movements have been reported to have significant effects on interhemispheric coherence in the gamma band as measured by EEG (*Propper et al., 2007*). But a more recent EEG study found scant evidence that eye movements altered interhemispheric coherence or that improvements in recall were correlated with changes in coherence (*Samara et al., 2011*).

## SIRE and handedness
A growing literature documents that SIRE is modulated by handedness. A study of both recall and recognition with strongly right-handed (SR) and non-strongly right handed
(nSR) subjects found that eye movements largely benefited the former while it was shown to be somewhat detrimental to the latter (*Lyle, Logan & Roediger 3rd, 2008*). This finding was initially interpreted to support the hemispheric interaction hypothesis. Better memory without eye movements is thought to exist in nSR individuals because nSR handedness is thought to be a behavioral marker for greater interhemispheric interaction. Consistent with this idea, middle-age nSR subjects perform better on tasks like paired associate recall and source memory, which likely depend on hemispheric interaction, compared to tasks like face recognition and forward digit span which depend less on hemispheric interaction (*Lyle, McCabe & Roediger, 2008*). For example, one study (*Lyle & Orsborn, 2011*) required subjects to classify faces as famous or novel with face presentation occurring in the left and right visual fields simultaneously (bilaterally) or in one field only (unilaterally). Famous faces were classified more quickly and accurately during bilateral presentation, reflecting that interhemispheric interaction facilitates famous face recognition, but neither inconsistent handedness nor saccades increased the size of bilateral gain.

Strong right-handedness, which is argued to be associated with decreased interhemispheric interaction, was associated with higher rates of false memories while bilateral saccades were associated with fewer false memories. Other evidence suggests that mixed-handers display better episodic memory in comparison with strong right-handers on assessments of explicit word recall and recall of real world events, but when corrected scores are analyzed handedness does not influence implicit word fragment completion (*Propper, Christman & Phaneuf, 2005*). Mixed-handedness and bilateral saccadic eye movements have also both been associated with an earlier offset of childhood amnesia and support the idea that interhemispheric interaction exerts effects on retrieval but not encoding of episodic memories. Mixed-handed subjects also demonstrate greater autobiographical recollection on components of seeing hearing and emotion compared to right-handed individuals, and 30 s of bilateral eye movements induce greater levels of autobiographical recollection across a range of components (*Parker & Dagnall, 2010*).

## From interhemispheric interaction to top down control

In a recent experiment in which subjects were shown arrays of lateralized letters and were asked whether target letters matched either of two probe letters, saccades were reported to enhance retrieval by increasing interaction among brain hemispheres (*Lyle & Martin, 2010*). Matching targets and probes were presented to either the same hemisphere or to separate hemispheres. Interhemispheric interaction was required on the across-hemisphere trials and intrahemispheric processing was required on the same hemisphere trials. Increased match detection accuracy was found on the within-hemisphere trials as a function of pre-task eye-movements suggesting saccades enhance intrahemispheric processing but not interhemispheric interaction. The nSR subjects showed higher across-hemisphere accuracy suggesting that an absence of strong right-handedness may reflect greater interhemispheric interaction. A recent comparison of consistent and inconsistent left- and right-handers on associative memory tests after saccade or no-saccade conditions showed that saccades enhanced retrieval for consistent-handers only, but impaired retrieval for inconsistent-handers (*Lyle et al., 2012*). This important study established

that handedness consistency, regardless of left or right direction, is an important factor to consider when studying memory.

Given the lack of convincing support for the idea that interhemispheric interaction underlies the bilateral saccade effects, more recent work has focused on the idea that saccade execution enhances cognition by altering attentional control. In one study, performance on the well-established revised attentional network test (*Fan et al., 2009*) was assessed after either repetitive bilateral saccades or central fixation (*Edlin & Lyle, 2013*). Saccade execution increased the executive function network, which encompasses attentional control, by decreasing response times to target stimuli in presence of response-incongruent flankers. In this study, the saccade-induced enhancement of attentional control occurred independently of handedness consistency. This raises the possibility that there could be a larger role for top-down attention when memories are more difficult to access, and recent results support that saccade execution has a greater facilitative effect on retrieval when recall and recognition are more difficult (*Lyle & Edlin, 2015*).

## Outstanding questions and current objectives

There remain several questions about the behavioral effects of repetitive saccades that have not yet been explored. The first is how do repetitive saccades affect memory for complex novel visual stimuli? The second involves how saccades affect stages of memory processing other than retrieval. Most previous investigations have had subjects perform bilateral saccades after encoding but before retrieval. It is not well understood how bilateral saccades affect the encoding of complex novel visual stimuli, nor is it understood how these effects are modulated by handedness. Understanding whether eye movements and handedness influence working memory or long-term recognition memory or both is also relevant to a decades-long theoretical debate concerning potential relationships between working memory and long-term memory. ''Buffer'' accounts (e.g., Baddeley's multi-component model) argue for a central executive coordinating separate phonological and visuo-spatial short-term stores that are separate from long-term memory (*Baddeley, 2003*), while the hierarchically arranged embedded process account argues working memory is a subset of activated memory in the focus of attention and the subset of long-term memory that is currently activated (*Cowan, 1999*). To address these questions, we conducted an exploratory study of the effects of repetitive bilateral saccades and handedness on scene memory. We utilized a variant of the well-known Sternberg working memory paradigm (*Sternberg, 1966*) in which during a series of trials five pre-experimentally novel scenes are presented and held online during a short 6 s delay period. After a 10-minute period of awake rest, we tested subjects' long-term recognition memory for scenes previously maintained in short-term working memory. We have shown previously that subjects perform this type of working memory task with scenes as stimuli and can recognize far above chance dozens of scenes shown during the working memory task after a long-term retention interval (*Ellmore et al., 2015*). Recent source analysis of scalp EEG acquired with this task has implicated a right hemisphere parieto-occipital region that is active when maintaining scenes in working memory, with temporal-spectral delay activity that correlates with subsequent probe memory retrieval (*Ellmore, Ng & Reichert, 2017*).

The specific objective of the present study was to test alternative hypotheses that making repetitive saccades before or after encoding novel scenes in working memory would have either beneficial or detrimental effects on subsequent memory as a function of handedness. A between-subjects design was employed that allowed us to test (1) whether making repetitive saccades *before* working memory encoding modulates subsequent working and long-term memory retrieval as a function of handedness and (2) whether making repetitive saccades *after* working memory encoding and before recognition modulates subsequent long-term recognition as a function of handedness.

## MATERIALS AND METHODS

### Subjects

A total of 111 subjects ($M = 20.74$, $SD =3.91$, range 18 to 40; 68 females, 43 males) were recruited through the Sona Systems online experimental scheduling system of the City College of New York Psychology Department. Each provided written informed consent, completed a revised Edinburgh Handedness Inventory (EHI) (*Williams, 2010*), and completed study procedures according to a protocol approved (approval number 2015-0550) by the Institutional Review Board of the City College of New York Human Research Protection Program. At the end of the study, each subject received extra course credit for a total of one hour of participation.

The target sample size of over 100 subjects was determined based on similar sample sizes used in previous published studies of eye movements and memory as well as on our ability to recruit as many participants as possible during the date range for study recruitment which began September 16, 2015 and ended November 22, 2017. In addition, for this study all measures, conditions, and data exclusions are reported herein.

### Experimental design

All subjects completed a working memory task, followed by a 10-minute period of quiet awake rest, followed lastly by a test of recognition memory. The study utilized a between-subjects design to manipulate eye movements by performance of two different eye tasks (saccade or fixation) executed either before performance of the working memory task (encoding group) or after the working memory task but before a test of recognition memory (recognition group). There was a total of 65 subjects ($M = 20.86$, $SD = 3.63$, range 18 to 38; 41 females, 24 males) in the encoding group and a total of 46 subjects ($M = 20.56$, $SD = 4.30$, range 18 to 40; 27 females, 19 males) in the recognition group. Within these two groups, subjects were assigned to perform either the saccade or fixation task using a pre-determined randomization schedule to minimize experimenter bias.

### Apparatus

All subjects completed tasks inside a closed room to minimize auditory and visual distractions. Tasks were programmed in SuperLab 5 (Cedrus Corporation, San Pedro, CA, USA). Tasks were displayed on a 28-inch LED monitor with a refresh rate of 60 Hertz and a screen resolution of 1,920 by 1,080 pixels. Participants sat in the dark 96 cm from the monitor on an adjustable stool and positioned their head in an Ultra Precision

Head Positioner™ (Arrington Research, Inc., Scottsdale, AZ, USA), a combined chin and forehead rest, to help minimize head movements. The experimenter remained in the room for the duration of the experiment, but was seated outside of subjects' view. A black curtain separated the experimenter from the subject to minimize distraction. To ensure subjects complied with instructions when required to make saccades or fixation, an infrared light emitter and a 220 Hz eye-tracking camera (*ViewPoint EyeTracker®* by Arrington Research, Inc; http://www.ArringtonResearch.com) was placed between the monitor and the subject to track the pupil of one eye. The experimenter monitored the *x* and *y* eye position as measured by real-time computation of pupil center-of-mass on a separate monitor only visible to the experimenter. Subject behavioral responses were collected using a Cedrus RB-740 response pad placed on the table in front of them.

## Tasks

### Working memory

Subjects completed 40 trials of a variant of a Sternberg working memory task (*Sternberg, 1966*) used in previous experiments (*Ellmore et al., 2015*; *Ellmore, Ng & Reichert, 2017*). Each trial consisted of a fixed load of five scene stimuli. A total of 220 pre-experimentally novel scenes were shown across the 40 trials of the working memory task. Each was a 24-bit color image of an outdoor scene sampled from the SUN database (*Xiao et al., 2010*). Scene stimuli were 800 by 600 pixels and were presented at the center of the monitor subtending a visual angle of 16 by 12 degrees. On each trial, five scenes appeared sequentially for 2 s each, followed by a 6 s delay period consisting of a white screen with a foveal crosshair, during which subjects were required to keep these scenes in mind, and then a probe scene was displayed for 2 s (Fig. 1A). Each trial was separated by 5 s of black screen. Half (20) of the trials contained a positive probe, which was one of the five scenes presented before the delay period. The other 20 trials contained a negative probe, a new scene not previously shown in the set of scenes presented before the delay period. Positive probes were randomly selected from each trial's five encoding stimuli so that the serial position of the probe was equally likely to come from encoding position 1, 2, 3, 4, or 5 across the set of WM trials. Positive and negative probe trials were randomly distributed across the set of 40 working memory trials. Upon the presentation of the probe in each working memory trial, subjects were instructed to press a green button on the response pad if the probe scene matched one of the scenes presented in the set of five scenes shown before the 6 s delay period; subjects were instructed to press a red button if the probe scene did not match one of the scenes presented in the set of five scenes shown before the delay. For each working memory probe presentation, subjects were required to respond within the 2 s that the probe was on the screen. Before performing the working memory task, subjects were given a 5-min demonstration task with different stimuli (animals) to ensure that they understood task instructions before the actual working memory task began.

## Recognition memory

After completing the working memory task, subjects were allowed to disengage from the head restraint and rest quietly in the experiment room for ten minutes. At the end of this ten-minute retention interval, subjects completed a task of recognition memory (Fig. 1B)
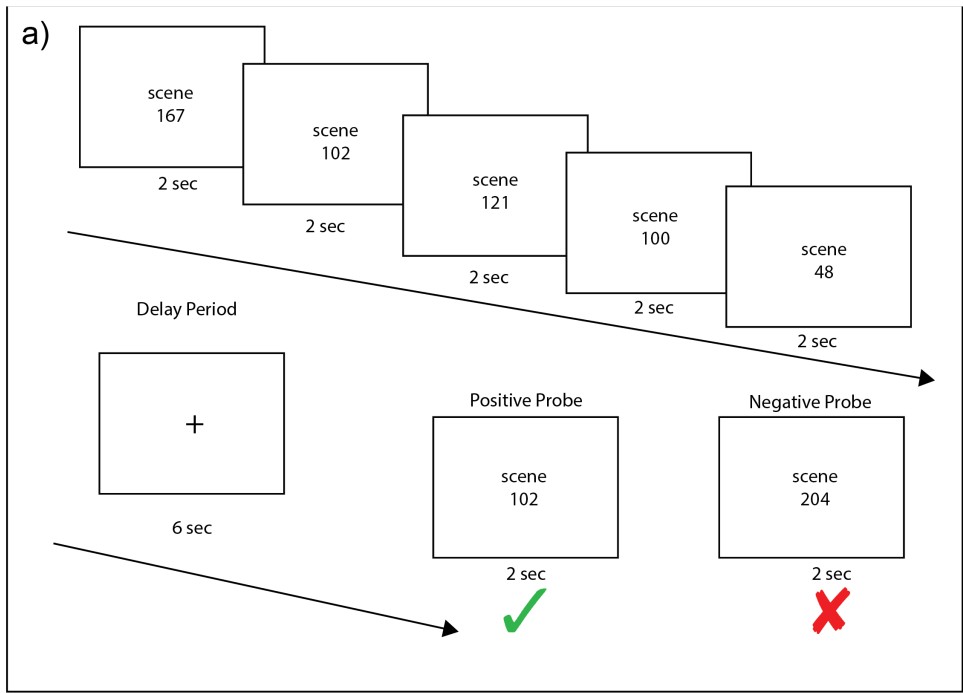

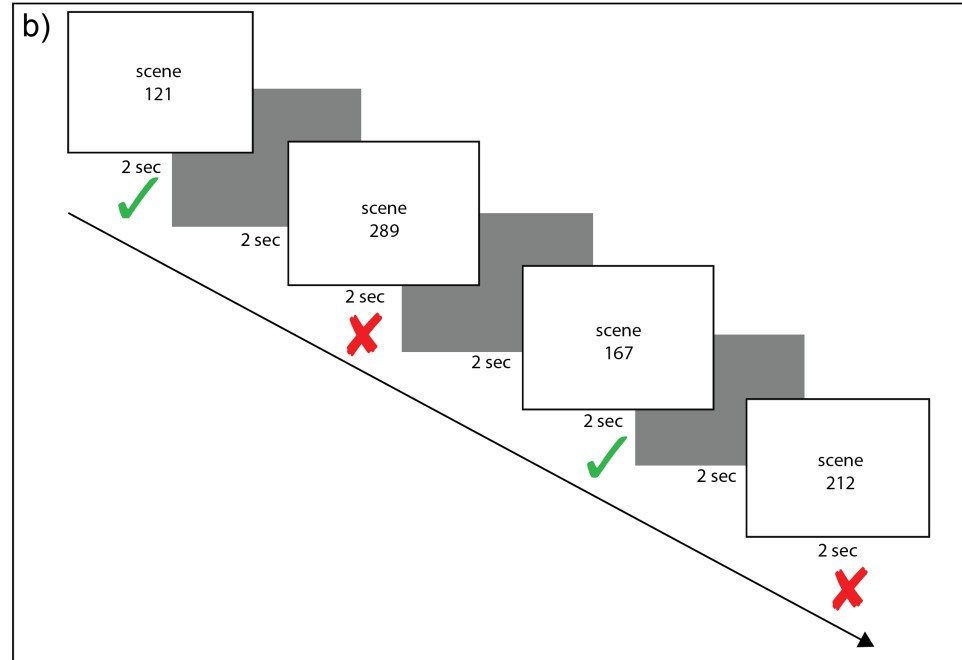

**Figure 1   Example scene working and recognition memory trials.** Each working memory trial (A) consisted of an encoding phase of five scene stimuli, each presented for 2 s, followed by a 6 s delay period of crosshair fixation and a 2 s presentation of either a positive probe (scene from the encoding set) or a negative probe (new scene). On each trial, there was a 50% chance of a positive probe appearing. Each recognition memory trial (B) consisted of alternating presentation of a scene stimuli for 2 s followed by a black screen for 2 s. Old and new scenes were randomly intermixed. 

**Figure 1 (…continued)**
Subjects were instructed to press a green button if they had seen a stimulus in any of the previous working memory trials and a red button if they had never seen the stimulus before. In this example, image numbers from the set of scene stimuli are listed rather than the actual scene images to illustrate how scenes during encoding could be presented as a positive probe (A) or as old stimuli in the recognition task (B).

that included a total of 200 scenes. One hundred of these scenes were randomly sampled from the sets of scenes presented before the delay in each of the 40 previous working memory trials. A total of eight out of these 100 "old" stimuli were previously used as positive probes in the WM task. The other 100 scenes were completely new scenes sampled randomly from the SUN database. The 100 old and 100 new scenes were presented one at a time for 2 s each in random order as individual trials. For each scene presentation during the recognition test, subjects were required to respond within the 2 s that the probe was on the screen. A 2 s black screen separated each trial. For each trial, subjects were instructed to press the green button on the response pad to indicate that they recognized having previously viewed the scene; subjects were instructed to press the red button to indicate that they did not recognize the scene.

## Eye tasks

The main experimental manipulation required subjects to perform one of two eye tasks (saccade or fixation) either before the working memory task or before the recognition memory task. The saccade task required subjects to make deliberate repetitive horizontal saccades for 30 s. Each horizontal saccade was cued by the appearance of a white disc on a black background. The diameter of the white disk was 70 pixels (1.49 degrees of visual angle). Each dot appeared for 500 ms and alternated its position between the right and left sides of the screen for a total of 30 s (Fig. 2A). The center-to-center distance of the disk as it alternated between the right and left sides of the screen was 1,285 pixels (27.3 degrees of visual angle). The fixation task required that subjects fixate a single disc (same size as the white disc in the saccade task) that was stationary at the center of the screen. The disc appeared on a black background for 30 s and cycled through six colors (red, blue, yellow, green, pink, and purple), with each color presented individually for 500 ms (Fig. 2C). To keep subjects' attention on the different eye tasks, subjects were instructed to count covertly the number of times the disc alternated positions between the left and right screen positions; during the fixation task, subjects were instructed to count covertly the number of times the disc changed colors. The number of disc alternations and color changes was equal.

## Analysis

The analysis began with the computation of a laterality quotient (LQ) from the responses to the eight questions of the EHI-R (*Williams, 2010*). The EHI-R (DOI 10.13140/RG.2. 2.33298.25284) is similar to the original EHI published in 1971 (*Oldfield, 1971*). Highly variable usage of modified versions of the original EHI are prominent in the scientific literature, which may imperil efforts to produce replicable and convergent findings (*Edlin et al., 2015*). Therefore, it is important to explain and scientifically justify differences in any

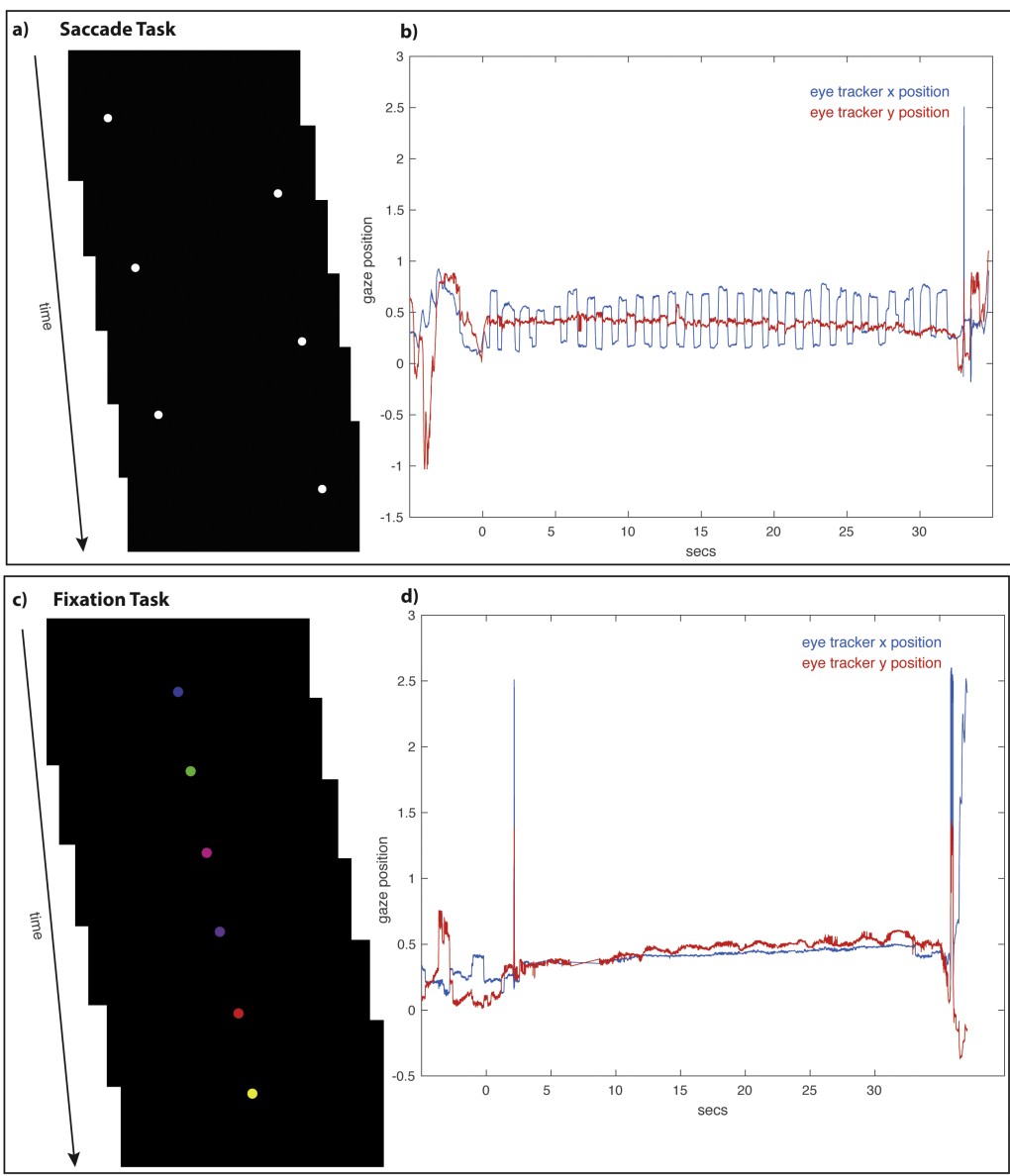

**Figure 2  The Saccade and Fixation Eye Tasks.** The saccade task (A) required subjects to alternate for 30 s looking toward the left and right as a white disc moved back and forth across (B) show saccadic movements with large periodic deviations in the *x* position (horizontal) trace with a stable *y* position (vertical) trace. The fixation task (C) required subjects to maintain fixation on a center disc as it changed color. Example eye-tracking traces (D) show fixation, with minimal deviations in the *x* (horizontal) and *y* position (vertical) traces.

revision and note differences with the inventory used in many other labs (*Lyle, McCabe & Roediger, 2008*). Compared to the original EHI, the EHI-R discards three items ("broom", "opening box", and "drawing") and adds a new one—"computer mouse"—making the total number of items eight. It also uses the Likert format in an attempt to simplify the confusing instructions of the original EHI. The items "opening box" and "broom" are

removed based on statistical evidence including factor analytic studies (*Dragovic, 2004*; *Williams, 1986*) that find a single Handedness factor whose loadings on it of these two activities are low outliers. "Drawing" is removed because it could be used as a substitute for writing since drawing and writing are very highly (about 0.9) correlated and the inclusion of both adds little new information. The addition of "computer mouse" is based on a suggestion by *Dragovic (2004)*. The use of a computer mouse is one major unimanual activity in today's world that was not widespread forty years ago when the EHI was developed. Additionally, the EHI-R has been used in several recent studies (*Eriksen et al., 2018*; *Holcombe, Nguyen & Goodbourn, 2017*; *Kielar et al., 2016*; *Gosser & Rice, 2015*). The EHI-R asks subject to indicate which hand they use to complete various everyday activities (e.g., striking a match, using a computer mouse). Participants indicated their handedness preference for each item by checking off one of the following response options (with corresponding point values): "always right" (50) "usually right" (25), "no preference" (0), "usually left" (−25), and "always left (−50). The LQ was calculated by tallying the point values for all eight items and dividing the total score by four, resulting in LQ scores that ranged from +100 (right handed) to −100 (left handed). A total of 7 subjects scored as left-handed (LQ < 0), one subject scored perfectly mixed (LQ = 0), 33 subjects scored non-strongly right-handed (LQ < 80), and 78 subjects scored strongly-right handed (LQ ≥ 80).

The next step in the analysis was to compute for each subject performance expressed as overall percentage correct and sensitivity as $d$-prime for the 40 trials of the working memory task and separately for the 200 trials of the recognition memory task. The $d$-prime sensitivity measure from signal detection theory accounts for effects of response bias (*Stanislaw & Todorov, 1999*). The proportions of total hits and false alarms were first calculated for each subject and each task. A hit was counted when a previously presented stimulus (i.e., a positive probe in the working memory task) was signaled by the subject pressing a button indicating, correctly, that the stimulus had been previously seen (an old stimulus correctly classified as old). A false alarm was counted when a (new) stimulus not previously presented (i.e., a negative probe in the working memory task) was indicated by the subject pressing a button indicating, incorrectly, that the stimulus had been previously presented (a new stimulus incorrectly classified as an old stimulus). Then $d$-prime was computed as the difference in standardized normal deviates of hits minus false alarms: *Z(hit rate) − Z(false alarm rate)*. Separate linear regression analyses were then performed with the criterion variables (Y) percent correct accuracy and $d$-prime sensitivity. The predictor variables (X) were raw LQ for direction of handedness and the absolute values of LQ for degree of handedness. Regressions were computed to predict the criterion variables of accuracy or $d$-prime by either direction or degree of handedness as a function of the eye-movement task (fixation vs. saccade) and placement of the eye-movement task (before the working memory or recognition memory task). Regression analyses were computed in GraphPad Prism version 7.0 (GraphPad Software, La Jolla California USA; http://www.graphpad.com). Statistical tests of the regression slopes were also computed to assess differences in performance as a function of laterality and type and placement of the eye-movement task. Statistical analysis of regression intercepts (elevations) was used

to assess differences in overall performance as a function of type and placement of the eye-movement task. The raw data input to these analyses is provided in the File S1.

## RESULTS

### Working memory performance
#### *Analysis of regression slopes*
Linear regression was used to quantify the relationship between working memory performance, as measured by both percent correct accuracy and $d$-prime, and direction of laterality as a function of the eye task. The statistical tests assessed whether the slope of the regression equation differed significantly from zero. Similar tests were used to assess the relationship between the dependent measures and the degree of laterality, indicating consistency of handedness, using the absolute value of the laterality quotient.

When fixation occurred before the working memory task, working memory percent correct accuracy increased non-significantly as a function of direction of laterality (slope = $0.03098 \pm 0.03765$ SE, $F_{(1,31)} = 0.6774$, $p = 0.4168$, Fig. 3A blue line and blue circles). The $d$-prime measure also increased non-significantly as a function of direction of laterality (slope = $0.00168 \pm 0.00337$ SE, $F_{(1,31)} = 0.2473$, $p = 0.6225$, Fig. 3B blue line and blue circles). Working memory percent correct accuracy increased non-significantly as a function of degree of laterality (slope = $0.02432 \pm 0.05178$ SE, $F_{(1,31)} = 0.2206$, $p = 0.6419$, Fig. 3C blue line and blue circles). The $d$-prime measure also increased non-significantly as a function of degree of laterality (slope = $0.001734 \pm 0.00461$ SE, $F_{(1,31)} = 0.141$, $p = 0.7098$, Fig. 3D blue line and blue circles).

When saccades occurred before the working memory task, working memory percent correct accuracy increased significantly as a function of direction of laterality (slope = $0.1029 \pm 0.04365$ SE, $F_{(1,30)} = 5.56$, $p = 0.0251$, Fig. 3A red line and red triangles). The $d$-prime measure increased non-significantly as a function of direction of laterality (slope = $0.0029 \pm 0.0032$ SE, $F_{(1,30)} = 0.8206$, $p = 0.3722$, Fig. 3B red line and red triangles). Working memory percent correct accuracy increased non-significantly as a function of degree of laterality (slope = $0.1212 \pm 0.1455$ SE, $F_{(1,30)} = 0.694$, $p = 0.4114$, Fig. 3C red line and red triangles). The $d$-prime measure also increased non-significantly as a function of degree of laterality (slope = $0.01137 \pm 0.00984$ SE, $F_{(1,30)} = 1.334$, $p = 0.2571$, Fig. 3D red line and red triangles).

The subjects who performed fixation and saccades after the working memory task serve as a control group, as one would expect no influence of these eye tasks on working memory performance if they occur after the working memory task. Consistent with this prediction, when fixation occurred after the working memory task, working memory percent correct accuracy increased non-significantly as a function of direction of laterality (slope = $0.02973 \pm 0.02872$ SE, $F_{(1,21)} = 1.072$, $p = 0.3123$, Fig. 3A grey line and grey boxes). The $d$-prime measure also increased non-significantly as a function of direction of laterality (slope = $0.00116 \pm 0.00268$ SE, $F_{(1,21)} = 0.1873$, $p = 0.6696$, Fig. 3B grey line and grey boxes). Working memory percent correct accuracy increased non-significantly as a function of degree of laterality (slope = $0.06811 \pm 0.05039$ SE, $F_{(1,21)} = 1.827$, $p = 0.1909$, Fig. 3C grey

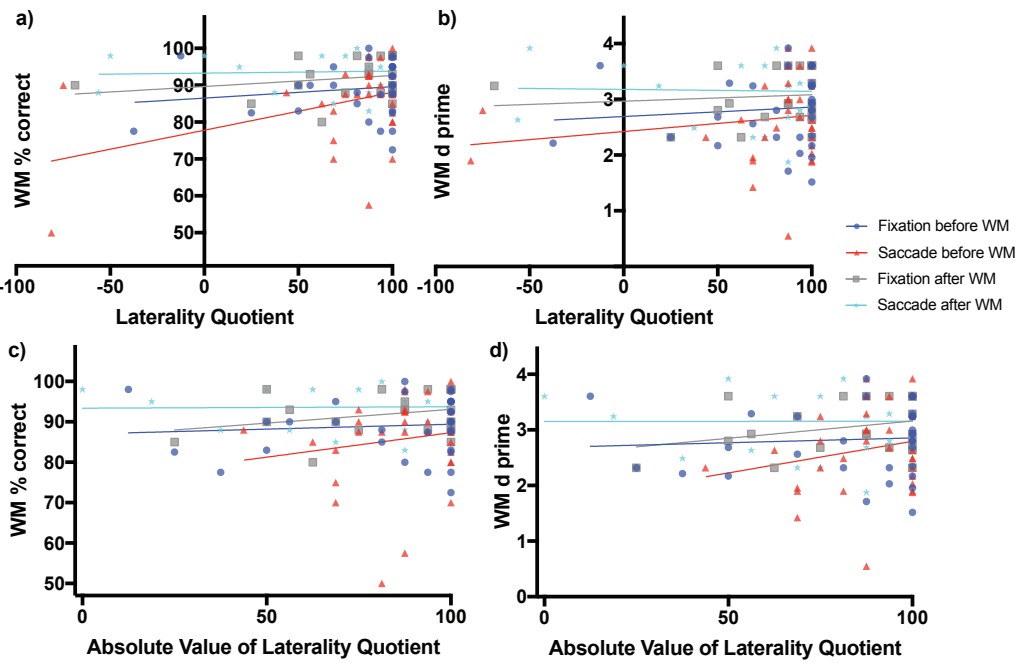

**Figure 3** **Scene working memory performance as a function of laterality and eye task timing.** There was a significant relationship in working memory percent correct performance as a function of direction of laterality when saccades were made before the working memory task (non-zero slope of $0.1029 \pm 0.04365$ SE, $F_{(1,30)} = 5.56$, $p = 0.0251$, red line and red triangles, A). Overall mean performance among conditions differed (non-zero intercepts, $F_{(3,106)} = 6.164$, $p = 0.0007$) with lowest working memory percent correct performance obtained when saccades were made before the working memory task ($77.79\% \pm 3.84$, 95% CI [69.95–85.64%]) and best performance when saccades were made after the working memory task ($93.28\% \pm 1.844$ SE, 95% CI [89.45–97.12]). (B) shows the relationship of the sensitivity measured $d$-prime with direction of laterality, while (C) and (D) show relationships between percent correct accuracy and $d$-prime respectively with degree of laterality as expressed by the absolute value of laterality quotients.

line and grey boxes). The $d$-prime measure also increased non-significantly as a function of degree of laterality (slope $= 0.00620 \pm 0.00462$ SE, $F_{(1,21)} = 1.805$, $p = 0.1935$, Fig. 3D grey line and grey boxes).

When saccades occurred after the working memory task, working memory percent correct accuracy increased non-significantly as a function of direction of laterality (slope $= 0.005429 \pm 0.02251$ SE, $F_{(1,21)} = 0.05818$, $p = 0.8117$, Fig. 3A cyan line and cyan stars). The $d$-prime measure decreased non-significantly as a function of direction of laterality (slope $= -0.00034 \pm 0.0025$ SE, $F_{(1,21)} = 0.0187$, $p = 0.8926$, Fig. 3B cyan line and cyan stars). Working memory percent correct accuracy increased non-significantly as a function of degree of laterality (slope $= 0.0035 \pm 0.03747$ SE, $F_{(1,21)} = 0.0087$, $p = 0.9265$, Fig. 3C cyan line and cyan stars). The $d$-prime measure also increased non-significantly as a function of degree of laterality (slope $= -6.445e-005 \pm 0.00422$ SE, $F_{(1,21)} = 0.00023$, $p = 0.9880$, Fig. 3D cyan line and cyan stars).

## Analysis of regression intercepts

While analysis of regression slopes indicates change in performance as a function of direction or degree of laterality and eye task timing, evaluation of the regression intercepts (i.e., elevations) indicates whether the overall levels of performance differ.

When considering relationships of performance with direction of laterality, the four intercepts of Fig. 3A were significantly non-zero ($F_{(3,106)} = 6.164$, $p = 0.0007$) with best working memory percent correct performance when saccades were made after the working memory task (93.28% $\pm$ 1.844 SE, 95% CI [89.45–97.12]) and worst working memory percent correct performance when saccades were made before the working memory task (77.79% $\pm$ 3.84, 95% CI [69.95–85.64%]). The four intercepts of Fig. 3B were also significantly non-zero ($F_{(3,106)} = 3.742$, $p = 0.0133$) with best working memory $d$-prime occurring when saccades were made after the working memory task (3.179 $\pm$ 0.2082 SE, 95% CI [2.746–3.612]) and worst working memory $d$-prime when saccades were made before the working memory task (2.442 $\pm$ 0.2819, 95% CI [1.847–2.998]).

When considering relationships of performance with degree of laterality, the four intercepts of Fig. 3C were significantly non-zero ($F_{(3,106)} = 5.699$, $p = 0.0012$) with best working memory percent correct performance when saccades were made after the working memory task (93.38% $\pm$ 3.07 SE, 95% CI [87.00–99.77]) and worst working memory percent correct performance when saccades were performed before the working memory task (75.2% $\pm$ 12.80, 95% [49.05–101.30%]). The four intercepts of Fig. 3D were also significantly non-zero ($F_{(3,106)} = 3.909$, $p = 0.0108$) with best working memory $d$-prime occurring when saccades were made after the working memory task (3.15 $\pm$ 0.3463 SE, 95% CI [2.43–3.87]) and worst working memory $d$-prime when saccades were made before the working memory task (1.658 $\pm$ 0.866, 95% CI [−0.1106–3.427]).

## Recognition performance
### Analysis of regression slopes

Linear regression was also used to quantify the relationship between recognition memory, as measured by both percent correct accuracy and $d$-prime, and direction of laterality as a function of the eye task. The statistical test assessed whether the slope of the regression equation differed significantly from zero. Similar tests were used to assess the relationship between the dependent measures and the degree of laterality, indicating consistency of handedness, using the absolute value of the laterality quotient.

When fixation occurred before the working memory task, later recognition percent correct accuracy increased non-significantly as a function of direction of laterality (slope = 0.03746 $\pm$ 0.04449 SE, $F_{(1,31)} = 0.7091$, $p = 0.4062$, Fig. 4A blue line and blue circles). The $d$-prime measure also increased non-significantly as a function of direction of laterality (slope = 0.00208 $\pm$ 0.00281 SE, $F_{(1,31)} = 1.351$, $p = 0.2542$, Fig. 4B blue line and blue circles). Recognition memory percent correct accuracy increased non-significantly as a function of degree of laterality (slope = 0.03984 $\pm$ 0.06102 SE, $F_{(1,31)} = 0.4263$, $p = 0.5186$, Fig. 4C blue line and blue circles). The $d$-prime measure also increased non-significantly as a function of degree of laterality (slope = 0.00344 $\pm$ 0.00383 SE, $F_{(1,31)} = 0.811$, $p = 0.3747$, Fig. 4D blue line and blue circles).
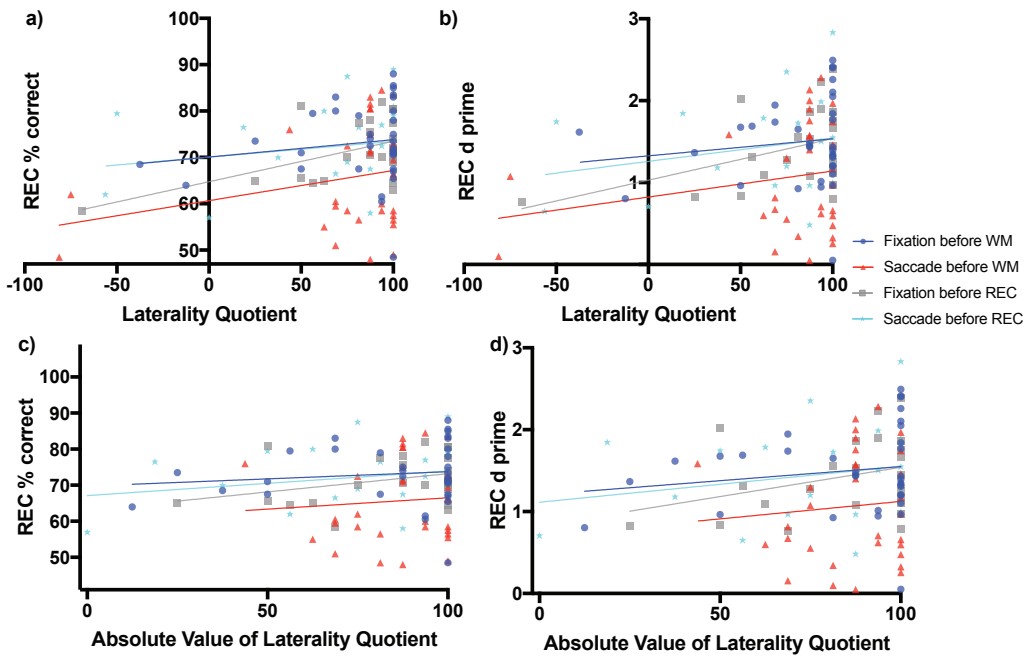

**Figure 4** **Scene Recognition Memory Performance as a Function of Laterality and Eye Task Timing.** There was a significant relationship in recognition percent correct performance as a function of direction of laterality during the condition of fixation before the recognition memory task (non-zero slope = $0.08738 \pm 0.03277$ SE, $F_{(1,21)} = 7.111$, $p = 0.0144$, grey line and grey boxes (A). Overall mean performance among the conditions differed ($F_{(3,106)} = 4.721$, $p = 0.0039$) with best recognition memory performance during the condition of fixation before the working memory task ($70.08\% \pm 3.899$ SE, 95% CI [62.12–78.03], blue line and blue circles, A) and lowest recognition memory performance during the condition when saccades were made before the working memory task ($60.67\% \pm 4.06$, 95% CI [52.38–68.96], red line and red triangles, A). (B) shows the relationship of the sensitivity measured $d$-prime with direction of laterality, while (C) and (D) show relationships between percent correct accuracy and $d$-prime respectively with degree of laterality as expressed by the absolute value of laterality quotients.

When saccades occurred before the working memory task, later recognition percent correct accuracy increased non-significantly as a function of direction of laterality (slope = $0.06502 \pm 0.04615$ SE, $F_{(1,30)} = 1.985$, $p = 0.1691$, Fig. 4A red line and red triangles). The $d$-prime measure for later recognition also increased non-significantly as a function of direction of laterality (slope = $0.0032 \pm 0.0027$ SE, $F_{(1,30)} = 1.351$, $p = 0.2542$, Fig. 4B red line and red triangles). Recognition memory percent correct accuracy also increased non-significantly as a function of degree of laterality (slope = $0.0629 \pm 0.1472$ SE, $F_{(1,30)} = 0.1829$, $p = 0.6720$, Fig. 4C red line and red triangles). The $d$-prime measure for later recognition also increased non-significantly as a function of degree of laterality (slope = $0.0043 \pm 0.00863$ SE, $F_{(1,30)} = 0.2443$, $p = 0.6247$, Fig. 4D red line and red triangles).

When fixation occurred after the working memory task and before the recognition memory task, recognition percent correct accuracy increased significantly as a function of direction of laterality (slope = $0.08738 \pm 0.03277$ SE, $F_{(1,21)} = 7.111$, $p = 0.0144$, Fig. 4A grey line and grey boxes). The $d$-prime measure for recognition increased marginally significantly as a function of direction of laterality (slope = $0.00507 \pm 0.00247$ SE,

$F_{(1,21)} = 4.23$, $p = 0.0523$, Fig. 4B grey line and grey boxes). Recognition memory percent correct accuracy increased non-significantly as a function of degree of laterality (slope = 0.1014 ± 0.0639 SE, $F_{(1,30)} = 2.517$, $p = 0.1276$, Fig. 4C grey line and grey triangles). The *d*-prime measure for recognition also increased non-significantly as a function of degree of laterality (slope = 0.00714 ± 0.00456 SE, $F_{(1,30)} = 2.449$, $p = 0.1326$, Fig. 4D grey line and grey triangles).

When saccades occurred after the working memory task and before the recognition memory task, recognition percent correct accuracy increased non-significantly as a function of laterality (slope = 0.03398 ± 0.03797 SE, $F_{(1,21)} = 0.8009$, $p = 0.3810$, Fig. 4A cyan line and cyan stars). The *d*-prime measure for recognition increased non-significantly as a function of direction of laterality (slope = 0.0028 ± 0.00253 SE, $F_{(1,21)} = 1.274$, $p = 0.2718$, Fig. 4B cyan line and cyan stars). Recognition memory percent correct accuracy increased non-significantly as a function of degree of laterality (slope = 0.0679 ± 0.0626 SE, $F_{(1,30)} = 1.18$, $p = 0.2897$, Fig. 4C cyan line and cyan stars). The *d*-prime measure for recognition also increased non-significantly as a function of degree of laterality (slope = 1.115 ± 0.3462 SE, $F_{(1,30)} = 1.076$, $p = 0.3113$, Fig. 4D cyan line and cyan stars).

## Analysis of regression intercepts

Analysis of regression intercepts was conducted to determine whether the overall level of recognition performance differed as a function of the eye task timing. The four intercepts of Fig. 4A representing recognition percent correct accuracy versus direction of laterality were significantly non-zero ($F_{(3,106)} = 4.721$, $p = 0.0039$) with best recognition memory performance for the condition of fixation occurring before the working memory task (70.08% ± 3.899 SE, 95% CI [62.12–78.03%], Fig. 4A blue line and blue circles) and the lowest recognition memory performance in the condition of saccades occurring before the working memory task (60.67% ± 4.06, 95% CI [52.38–68.96%], Fig. 4A red line and red triangles). Recognition performance during the condition when saccades were made before the recognition memory task (70.06% ± 3.111, 95% CI [63.59–76.53], Fig. 4A cyan line and cyan stars) was greater than performance during the condition of fixation before the recognition memory task (64.75% ± 2.819, 95% CI [58.38–70.62%], Fig. 4A grey line and grey boxes), but the difference was not significantly different (two-tailed $t_{(44)} = 1.2648$, $p = 0.2126$, Cohen's $d = 0.3729$). The level of recognition performance in the condition when saccades were made before the recognition task was nearly identical to performance after the condition of fixation before the working memory task (70.08% ± 3.899, 95% CI [62.12–78.03%], Fig. 4A blue line and blue circles). These virtually identical levels of performance make the blue and cyan lines of Fig. 4A appear to overlap.

The four intercepts of Fig. 4B representing *d*-prime versus direction of laterality were significantly non-zero ($F_{(3,106)} = 3.964$, $p = 0.0101$) with best recognition memory performance for the condition of fixation occurring before the working memory task (1.327 ± 0.2467 SE, 95% CI [0.8235–1.83], Fig. 4B blue line and blue circles) and the lowest recognition memory performance in the condition of saccades occurring before the working memory task (0.8231 ± 0.2408, 95% CI [0.3313–1.315], Fig. 4B red line and red triangles).

The four intercepts of Fig. 4C representing recognition percent correct accuracy versus degree of laterality were significantly non-zero ($F_{(3,106)} = 4.863$, $p = 0.0033$) with best recognition memory performance for the condition of fixation occurring before the working memory task (69.76% ± 5.348 SE, 95% CI [58.86–80.67], Fig. 4C blue line and blue circles) and the lowest recognition memory performance in the condition of saccades occurring before the working memory task (60.22 ± 12.95, 95% CI [33.78–86.66], Fig. 4C red line and red triangles).

The four intercepts of Fig. 4D representing recognition $d$-prime versus degree of laterality were significantly non-zero ($F_{(3,106)} = 4.235$, $p = 0.0072$) with best recognition memory performance for the condition of fixation occurring before the working memory task (1.205 ± 0.3354 SE, 95% CI [0.5211–1.889], Fig. 4D blue line and blue circles) and the lowest recognition memory performance in the condition of saccades occurring before the working memory task (0.6978 ± 0.7593, 95% CI [−0.853–2.249], Fig. 4D red line and red triangles).

## DISCUSSION

Humans have a remarkable ability to remember a large number of scenes and other complex visual stimuli even when given only a short period of time to encode them (*Brown & Scott, 1971*; *Shepard, 1967*; *Standing, 1973*; *Standing, Conezio & Haber, 1970*). Scene memory is also an important component of episodic autobiographical memory. Normal as well as disordered reminiscence often includes vivid mental recollection of scenes where past events took place (*Brewin, 2014*; *Burgess, Maguire & O'Keefe, 2002*). In addition, recognition of a previously encountered scene in, for example an old photograph, is often quick, highly accurate, and can trigger episodic memory retrieval. Therefore, it is important to understand how scene memory is influenced by multifactorial traits like handedness and by simple behavioral manipulations (e.g., bilateral saccades) shown previously to influence episodic memory retrieval.

The present study was conducted to understand how both short- and long-term scene memory is modulated by handedness and by bilateral horizontal saccades made before scene encoding and recognition. Previous studies have shown that strongly right-handed and consistently-handed benefit most from bilateral saccades (*Chu, Abeare & Bondy, 2012*; *Lyle et al., 2012*; *Lyle & Orsborn, 2011*; *Prichard & Christman, 2017*; *Propper et al., 2017*), but these studies focused on mostly verbal aspects of explicit or episodic memory retrieval and had subjects make the saccades after encoding and before retrieval. The two novel aspects of the present study include, first, the focus on visual scene memory and, second, having different groups of subjects make saccades before encoding as well as before recognition.

There are two main findings of the present study. First, when saccades were made *before* encoding scenes in working memory, performance as measured by regression slopes relating overall percent correct with direction of handedness increased significantly positively (Fig. 3A) with right-handers (+LQ) performing better than left-handers (−LQ). Regression slopes exhibited the same trend of increasing magnitude when a $d$-prime measure of sensitivity was related to raw LQ scores (Fig. 3B), but there was no significant

relationship suggesting some effects of response bias may have been present in the subjects when a measure of overall percent correct accuracy was used. Regression slopes were mostly positive when percent correct and *d*-prime were regressed against absolute LQ, a measure of degree or consistency of handedness, but these tests also did not reach significance. Performance also did not correlate significantly as a function of direction of handedness when either the saccade or fixation condition came *after* the scene working memory task.

The second main result was that there was no apparent benefit of direction or degree of handedness on making saccades before scene recognition. Recognition performance in the group of subjects who made saccades after working memory and before recognition increased non-significantly as a function of both direction and degree of handedness. Recognition performance as measured by both overall percent correct and *d*-prime in the group of subjects who fixated after working memory and before recognition did increase significantly as a function of direction of handedness, with better performance as LQ increased indicating an advantage for right-handed subjects. The relationships between recognition performance and degree of handedness in those who fixated after working memory and before recognition were non-significantly positive. Finally, the subjects who made saccades before the working memory task performed the poorest during a subsequent test of recognition memory suggesting a link between poor encoding in working memory and poor subsequent long-term recognition memory.

Why would making saccades before encoding scenes in working memory result in better performance for strongly right-handed (+LQ) but not necessarily for consistent-handed individuals? There is some neuroimaging evidence suggesting that form-specific perceptual aspects of scene encoding may be right lateralized in the brain (*Brewer et al., 1998*; *Stevens et al., 2012*), with form-abstract and language-specific aspects including the verbalizability of the visual stimuli lateralized to the left (*Golby et al., 2001*; *Stevens et al., 2012*). Our recent source analysis of working memory delay period EEG implicates right parieto-occipital focus of activity that covaries positively as a function of performance. Due to limited sample size in that study, it's not possible to draw conclusions about whether mixed- or left-handers have a more symmetrical pattern of activity during scene encoding and memory maintenance that correlates with worse performance. However, a tentative explanation is that strongly right-handed individuals may have an advantage over mixed- and left-handers in controlling spatial attention for scene encoding and memory. Right handers perform better than either mixed or left handers on tasks requiring reproduction from memory of some aspects of geometric visual material, while left and mixed handers do not differ from one another (*Nebes & Briggs, 1974*). A hemispheric advantage for directing spatial attention could protect these right-handers from potentially destabilizing effects of repetitive saccades on visuospatial attention processes needed to encode the perceptual aspects of scenes. While saccades may be beneficial to mixed/inconsistent-handers before the retrieval of explicit episodic memories by improving attentional control—a form of top-down inward control—they may be detrimental to control of visuospatial attention directed outward to maintain a stable reference frame for encoding details of complex visual information contained in scenes. Fitting with this idea, there is evidence for hemispheric specialization at early levels of visual analysis to the right for processing low spatial

frequencies and to the left for processing high spatial frequencies, with supplementary activation of right inferior parietal lobule reflecting attentional modulation (*Peyrin et al., 2004*).

A growing number of studies have demonstrated saccade induced benefits specifically on recognition (*Brunye et al., 2009*; *Lyle & Orsborn, 2011*; *Parker & Dagnall, 2007*; *Parker, Relph & Dagnall, 2008*). In the present study, when bilateral saccades were made before scene recognition but not before scene working memory, we found a small but not statistically significant increase in performance relative to simple fixation. Performance was best for consistent right-handers, which includes those who have an LQ near +100 in Figs. 3 and 4 because they indicate they perform all tasks "always" with their right hand. A finding reported recently is that beneficial effects of pre-task saccades occur in consistent-handed individuals (*Lyle et al., 2012*), those who perform most actions consistently with either their right or their left hand. If this pattern was present in our data, we might see an inverted U-shaped distribution of data points in Fig. 4 with lower baseline performance for both −LQ (consistent left-handers) and +LQ (consistent right-handers) relative to mixed-handers. The ends of the inverted U-shaped might then be shifted up and/or flattened relative to fixation following saccades. Benefits are not seen in Fig. 4 for both left and right consistent-handed individuals. Rather, it appears that only consistent right-handers are better able to deal with the detrimental effects of bilateral saccades made before scene working memory (as indicated by the significant slope of the red line in Fig. 3A). Consistent right-handers also have better recognition memory in general because when they fixate before the recognition test they perform better (as indicated by the significant slope of the grey line in Fig. 4A).

The result of the present study represents a novel amendment to the important findings of previous studies showing better overall memory in inconsistent handers and beneficial effects of saccades in consistent-handers. Those findings have replicated in more recent studies (*Chu, Abeare & Bondy, 2012*; *Parker, Parkin & Dagnall, 2017*; *Prichard & Christman, 2017*; *Propper et al., 2017*). Many of the tasks used to assess episodic and/or autobiographical memories operate in the verbal domain. In the present study, subjects might default to a familiarity-based and highly-visual implicit "know" strategy rather than rely on a verbal episodic-like type of remembering. Adopting a familiarity-based strategy in the present experiments is even more likely given the speeded choices subjects were required to make within the 2 s stimulus viewing windows. Eye movement effects have been demonstrated to be more beneficial for remember/recollective responses or for tasks that demand recollection for accurate responses rather than familiarity (*Lyle et al., 2012*; *Parker, Relph & Dagnall, 2008*). In addition, many recognition tasks shown to benefit from bilateral saccades rely on word lists for memoranda (*Parker & Dagnall, 2007*; *Propper et al., 2017*; *Samara et al., 2011*), content that is much different from the novel scene stimuli used in the present study. However, it is also worth noting that the beneficial effects of saccades do not manifest in implicit memory tasks even when those tasks involve words. *Christman et al. (2003)* had participants study a short word list and then tested them on a word-stem completion task, consisting of an equal mix of new and previously seen words. The findings showed that none of the eye movement conditions enhanced word fragment

completion performance compared to the fixation condition. This was interpreted to indicate that saccades do not enhance implicit forms of memory that are likely reflective of intra-hemispheric activity (*Christman et al., 2003*).

One recent study of saccade related enhancement is notable for its use of visual-spatial stimuli and an exploration of whether effects could be obtained when eye movements were made before encoding. *Brunye et al. (2009)* used an encoding task that lasted a total of only 120 s and comprised a small number of just four stimuli, all aerial maps that were acquired from satellite pictures. The authors found increased recognition sensitivity and decreased response times with effects only seen when eye movements preceded episodic memory retrieval, but not when they preceded encoding. The effects were strongest for strongly right-handed individuals. On the basis of the HERA model (*Nyberg, Cabeza & Tulving, 1996*), Brunye et al. theorized that eye movements would not benefit encoding due to unilateral activity, but would be beneficial with recognition tests demanding a high degree of right and left-hemisphere activity.

The present study has some limitations. One limitation is the few strongly left-handed subjects in our sample. Although we found that when saccades occurred before the working memory task the WM percent correct accuracy increased significantly as a function of direction of laterality, this result could have been driven by the single very low performing left-hander represented by the red triangle in the lower left corner of Fig. 3A. The few left-handed subjects and large number of strongly-right handed subjects in the sample rules out a traditional analysis to assess group-level differences. Instead in the present study relative performance differences as a function of LQ were characterized using regression slopes. Therefore, summaries of results involving right-handers (+LQ) performing better than left-handers (−LQ) are not intended and should not be taken to imply there were group-level differences between equal samples of left- and right-handers. Another limitation is that the working memory task utilized a relatively high load of 5 scene stimuli per trial and required a large number of scenes to be memorized and later discriminated in the recognition test. The 200-trial recognition test took over thirteen minutes to complete and it is possible that if there were any beneficial effects of preceding eye movements they may have reduced across the lengthy time interval taken for subjects to complete the task. This increased the overall difficulty of the memory tasks with performance during the working memory task nowhere near ceiling. Prior studies involving saccade manipulation have used relatively smaller numbers of items for encoding and have frequently required participants to view stimuli passively during learning, rather than engage them in working memory. Both the limited number of items and the nature of the encoding tasks used in previous research require a lower degree of cognitive processing compared to the high-load working memory and recognition tasks used here. Both tasks far exceeded the numbers of stimuli used in prior studies and could have potentially decreased overall encoding and consolidation into long-term memory given the short 10-min interval separating working memory and recognition testing. For the eye tasks, the requirement for subjects to count disc alternations and color changes was to ensure subjects performed the tasks. However, an important difference to acknowledge is that counting has not commonly been used in previous studies of saccade effects on memory. While our sample included 111

participants, there was an unequal sample of strongly left-handed compared to strongly-right handed subjects making the degree of handedness (i.e., consistency comparisons) weighted unequally with strongly-right handed subjects. Finally, the study design relied on between-subjects manipulation of conditions, which tends to increase variance and decrease statistical power relative to a within-subject design.

## CONCLUSION

We found evidence that performance of repetitive horizontal saccades before scene working and recognition memory affects performance differently depending on the direction but not degree of handedness. Making repetitive saccades before encoding scenes in working memory has detrimental effects on both short- and long-term memory, with the most pronounced effects for those who are not strongly right-handed. Moreover, saccade execution before scene recognition does not appear to benefit recognition accuracy appreciably. Making repetitive saccades before scene encoding may be detrimental because the eye movements destabilize subsequent visuospatial processing. If working and long-term memory systems interact, then scenes that are poorly encoded initially will not transfer to long-term memory. Right handers may have an advantage in dealing with such disruption due to more lateralized mechanisms for scene processing. These tentative explanations require confirmation by future experiments combining both behavior and brain measurements.

## ACKNOWLEDGEMENTS

The authors thank Iffat Noor and Chelsea Reichert for assistance with data collection. The content is solely the responsibility of the authors and does not necessarily represent the official views of the National Institutes of Health.

### Funding

This work was supported by PSC-CUNY award ENHC-46-25 (TME). Research reported in this publication was supported by the National Institute of General Medical Sciences of the National Institutes of Health under Award Number SC2GM109346 (TME). The funders had no role in study design, data collection and analysis, decision to publish, or preparation of the manuscript.

### Grant Disclosures

The following grant information was disclosed by the authors:
PSC-CUNY: ENHC-46-25.
National Institute of General Medical Sciences of the National Institutes of Health: SC2GM109346.

### Competing Interests

The authors declare there are no competing interests.

## Author Contributions

- Timothy M. Ellmore conceived and designed the experiments, analyzed the data, prepared figures and/or tables, authored or reviewed drafts of the paper, approved the final draft.
- Bridget Mackin conceived and designed the experiments, performed the experiments, analyzed the data, approved the final draft.
- Kenneth Ng performed the experiments, analyzed the data, authored or reviewed drafts of the paper, approved the final draft.

## Human Ethics

The following information was supplied relating to ethical approvals (i.e., approving body and any reference numbers):

The Institutional Review Board of the City College of New York Human Research Protection Program (approval number 2015-0550) approved this study.

## Data Availability

Measurements for individual subjects that were input into the regression analyses are provided as comma-separate values in the Supplemental File.

## Supplemental Information

Supplemental information for this article can be found online at http://dx.doi.org/10.7717/peerj.5969#supplemental-information.

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
