# Peer review of "Saccades and handedness interact to affect scene memory"

_PeerJ, doi:10.7717/peerj.5969_

## Round 0.1 · original submission · Major Revisions

I have received reviews from three experts in the field. As you will see, they tend to be in agreement about the strengths and weaknesses of your manuscript and provide clear advice on how it can be improved.

Beyond their suggestions, I have only one item I would like you to add to the manuscript before resubmitting. I request that you add a statement to the paper confirming whether, for all experiments, you have reported all measures, conditions, data exclusions, and how you determined your sample sizes. You should, of course, add any additional text to ensure the statement is accurate. This is the standard reviewer disclosure request endorsed by the Center for Open Science [see http://osf.io/project/hadz3]. I include it in every review.

·

Basic reporting

The authors present the results of a single experiment that investigates the effects of both handedness and eye-movements on a STM and LTM task. It was found that when EMs were made prior to the STM task, then a relationship between handedness and performance was observed. However, EMs prior to recognition testing had no effect and was unrelated to handedness. This experiment was well conducted and explored some important issues. I have some comments to make that I think could be addressed in a revision. I have placed these below on a section-by-section basis.

Introduction
The authors provide a good and wide-ranging introduction that covers relevant existing research on SIRE and handedness effects on memory. Although I believe this to be good, I would like to suggest some revisions.

1. Line 67. I think the sentence could be modified slightly – thus:
“Strong right-handedness, which is associated with decreased interhemispheric interaction, . . .”
TO
“Strong right-handedness, which is argued to be associated with decreased interhemispheric interaction, . . .”
I say this in order to indicate the provisional nature of the claim.

2. Line 71. The authors appear to indicate that mixed handedness is associated with superior implicit memory as measured by WFC. In actual fact, if the corrected scores are analysed then handedness did not influence WFC (Propper eta l., 2005).

3. The sentences covering line 82 to about 86. I think this could be revised and made clearer – thus:
“A study of both recall and recognition with strongly right-handed (SR) and non-strongly right handed (nSR) subjects found that eye movements made little difference or could even be detrimental in nSR subjects. This finding supports the hemispheric interaction hypothesis but with the caveat that eye movements could be harmful for nSR individuals (Lyle et al. 2008a).
TO
“A study of both recall and recognition that compared strongly right-handed (SR) to and non-strongly right handed (nSR) subjects found that eye movements largely benefited the former whilst was shown to be somewhat detrimental to the latter (Lyle et al. 2008a). This finding was taken to support the hemispheric interaction hypothesis

4. Lines 109-111. The authors mention a recognition task that demands a high degree of right-left activity. Can I suggest that they indicate that this was old-new recognition and was compared to forced-choice recognition. For the former, they could say that this purportedly depends on right-left interaction.

5. For the final part of the introduction the authors note that the present concern is with both working memory and long-term recognition memory. I think that at this juncture it would be good to elucidate to the reader the potential relationships that could exist between WM and LTM and then consider this from the perspective of SIRE/handedness. I appreciate that this work is exploratory but I think it could be useful. This is just a suggestion for the authors and it could also be later reflected upon in the discussion.

6. As I noted, coverage of relevant material is good but I think that the introduction could benefit from a little restructuring in order to improve the presentation of the background work and their ideas.

For example - maybe consider using labelled subsections and (i) providing separate parts on SIRE and handedness effects prior to considering them together (ii), providing a separate section dealing with various problems of the cited work and the hemispheric interaction account, (iii) introduce the top-down control ideas and how this is related to certain experiment findings such as Lyle & Martin (2010) and Lyle & Edlin (2015).

Experimental design

Here I have commented upon the methods and results.

Method
1. I think that the design section could be presented in a clearer and more concise manner. For example: “The predictor variables were the eye-movement task (fixation vs. saccade) and placement of the EM task (before WM or before recognition). Both were between-subjects and the number/descriptives of the subjects in each group were . . . etc. The criterion variables were . . . etc
I used the terms predictor and criterion variables here given that regression analyses were performed.
2. What was the degree of visual angle for the EM task?
3. For the recognition task, were subjects required to respond with the 2 seconds that the picture was on the screen?

Results
Was there any rationale for using regression for the analyses? Previous work has always used ANOVA. Was this to avoid treating the handedness score as a dichotomous variable?
Perhaps the authors could also indicate why only some of the DVs were explored in the presented analyses.

Validity of the findings

The results are evaluated in a thorough manner and relate clearly to the results and findings. The conclusions follow logically from the findings. I have just a few points that relate to the discussion.

Line 449. At this point the authors note that the subjects may have used a familiarity-based strategy to perform the recognition test. They could cite work that has shown EM effects to be more beneficial for remember/recollective responses or tasks that demand recollection for accurate responses rather than familiarity (e.g., Parker, Relph & Daganall 2008 and Lyle, Hanaver-Torrez, Hacklander, and Edlin 2012).

In relation to the above, if subjects were required to response rapidly (e.g., within a 2 second window) then this might have fostered the use of a familiarity-based strategy.

Line 457. Small suggestion = perhaps the sentence could be reworded
“The findings showed that neither the eye movement nor fixation conditions elicited significant improvement, . .”
TO something like
“The findings showed that none of the eye movement conditions enhanced WFC performance compared to the fixation condition” This was taken to indicate saccades do not enhance implicit forms of memory that are likely reflective of intra-hemispheric activity (Christman et al. 2003).

For the recognition task, each picture appeared on the screen for 2 s followed by an ISI of 2 s. As there were 200 trials, this equals 800 secs overs = 13. 33 mins. This is quite a long time and I wonder if the effects of the EMs may have reduced across this time interval. Perhaps worth a discussion point?

Additional comments

Overall, some interesting issues were explored in this experiment. It’s a pity that the EM manipulation didn’t influence recognition, but maybe this is an example of a boundary condition for SIRE effects. I have indicated a major revision - but please do not let that put you off. The main reason I clicked on this option was simply related to the range of comments I have made (NOTE - many of these are very small indeed and doable in next to no time).
I believe the revision should not be in any way onerous and I reckon the authors could do it rather easily and in a very short space of time!! The largest job (should the authors agree) - relates to some restructuring/reorganisation in the introduction.

·

Basic reporting

I believe this manuscript would benefit from a thorough rewriting. The authors clearly wish to interpret effects of saccade execution memory in terms of interhemispheric interaction. This approach is outdated. In reviewing the literature, the authors do not provide a single piece of evidence supporting the idea that saccade execution affects memory (or any other form of cognition) by increasing interhemispheric interaction. On the contrary, the authors cite evidence that weighs AGAINST the idea that interhemispheric interaction is at the root of saccade-execution effects (lines 93-96, 117-119, and 129-131). Nonetheless, the authors' primary conclusion (lines 490-491 and also stated in the abstract) is that scene processing does not depend on interhemispheric interaction and this is why they failed to find saccade-induced retrieval enhancement. It is difficult to see how this conclusion follows from what has come before. I would also point out that this conclusion says nothing about why saccade execution HARMED scene memory. I would encourage the authors to think about what scene processing DOES depend on (versus focusing on what it DOESN'T depend on) and then consider why saccade execution HARMED scene memory. The authors have some potentially interesting findings here but I don't think they do them justice.
Another important consideration is that prior research has been primarily concerned with differences between inconsistent and consistent individuals. The authors, in contrast, seem more interested in strong right-handedness versus its absence (see, for example, statements made in the abstract). I myself was originally interested in the strong right-handed versus not distinction but the literature has advanced past that point.

Experimental design

I have no serious concerns about the experimental design but I think the authors must do a better job of acknowledging how their methodology differs from standard practice in studies of handedness and saccade execution. One important difference is that the authors used a different handedness inventory than is used in the labs that have generated most of the research that the authors cite. I am thinking of Christman's lab, Propper's lab, Parker and Dagnall's lab, and my lab. The inventory used in those labs is described in Lyle, McCabe, and Roediger (2008). I am not suggesting that the authors' use of a different inventory invalidates their findings. I have nothing of the sort in mind. I am only saying that the authors should acknowledge the difference and discuss its implications. Also, the authors do not report the visual angle subtended by the left and right discs in the eye-movement task. This is critical information. The labs mentioned above have all used the same visual angle. Furthermore, the authors had subjects count the number of times the disc changed positions or colors. To my knowledge, that requirement has not been imposed in any previous studies. Again, I think the authors must acknowledge this difference, as they acknowledge other important differences (lines 471-478).

Validity of the findings

No comment.

Additional comments

My comments under Basic Reporting and Experimental Design will no doubt be discouraging for the authors. I therefore wish to be clear that I think the authors have the basis of a potentially publishable paper here.

·

Basic reporting

- Lines 97-101: citations should be provided for points 1 and 2

- Lines 129-131: the authors should cite a study by Propper et al. that found that eye movements had significant effects on interhemispheric coherence in the gamma band:

Propper, R.E., Pierce, J., Bellorado, N., Geisler, M.W., & Christman, S.D. (2007). Effects of bilateral eye movements on interhemispheric gamma EEG coherence: Implications for EMDR therapy. Journal of Nervous and Mental Disease, 95, 785-788.

- Lines 146: the authors may be interested in citing an unpublished conference presentation that showed that eye movements before encoding hurt subsequent memory performance:

Christman, S.D., & Butler, M. (2005). Bilateral eye movements impair the encoding and enhance the retrieval of episodic memories. Presented at the 46th Annual Meeting of the Psychonomic Society, Toronto.

- Line 151: it is not clear what “a fixed load of 5” refers to. Five what?

- Lines 465-467: what were the results of the Brunye et al. (2009) study?

Experimental design

- Lines 204-206: given that participants’ eye movements were monitored by the experimenter, it is not clear why the details about eye tracking equipment were provided in the previous paragraph

Validity of the findings

This manuscript presents a single experiment looking at how eye movements and handedness interact in influencing working memory and long-term recognition memory for complex natural scenes. The two most valuable aspects of the study is (i) extending the meagre existing literature on how eye movements affect different handedness groups differently, and (ii) extending this paradigm to nonverbal stimuli.

My biggest concern centers on the dependent variables used to assess memory performance. It appears that the authors are simply using percent correct as their d.v. However, such a procedure confounds accuracy with response bias. To illustrate an extreme case, if a subject in their study responded that they recognized every single stimulus, that subject would have a 100% hit rate but also a 100% false alarm rate. To control for this, it is imperative that the authors use some sort of corrected accuracy score. I highly recommend performing signal detection analyses, allowing for independent examination of accuracy (d’) versus response bias (beta). However, the simpler “Hits minus False Alarms” might be sufficient. As it stands now, it is impossible to precisely determine the effects of eye movements and handedness on memory accuracy because the authors’ current analyses make it impossible to examine the effects of response bias.

The authors also need to provide more information about the revised version of the Edinburgh Handedness Inventory that they used. They provide a citation to Williams, 2010. However, the citation info in the reference section is incomplete, lacking a journal title, volume number, and pages. Being a very active handedness researcher, I was surprised that I was not aware of this Williams study; the authors need to provide the full citation info. I was particularly concerned by one of the two sample items mentioned by the authors: “using a computer mouse”. I have never seen this item on the EHI before, and need more details about how the “EHI-R” used by the authors was constructed and validated. I recommend the authors read the following article on EHI revisions:

Edlin, J.M., Leppanen, M.L., Fain, R.J., Hanaver-Torrez, S.D., & Lyle, K.B. (2015). On the use (and misuse?) of the Edinburgh Handedness Inventory. Brain and Cognition, 94, 44-51.

Another issue involving handedness is how the EHI-R scores were used. The authors used the raw EHI-R scores in their regression analyses. However, given the growing emphasis on degree of handedness (strong/consistent versus mixed/inconsistent), many recent handedness studies convert raw EHI scores into absolute values. This gives a scale that ranges from mixed to strong (instead of from left to right). I highly recommend that the authors perform additional regression analyses on absolute value EHI-R scores.

---

## Round 0.2 · Minor Revisions

I have received reviews from the original three reviewers on the first manuscript. All are positive about the manuscript, but there are some lingering revisions that they have suggested. They should be easily addressed and I look forward to receiving a revised version of the document.

·

Basic reporting

I would like to thank the authors for taking on board the revisions based on my previous set of comments. Overall, I think they have done a commendable job and the paper reads much better and is structed in a more coherent manner compared to the first submission. In this context I have just a small range of minor points for the authors to consider.

Minor Points:

Line 60: After “everyday events” there should not be a full stop before the Christman citation.

Line 61: Maybe just acknowledge the first use of the term SIRE by Lyle and Colleagues.

Line 72: Part of this sentence is incorrect. We found effects for horizontal eye-movements not vertical.

Lines 131- 132: I have just suggested a slight change to a sentence to make the description of the findings more precise.

Thus, rather than:
“Increased match detection accuracy was found on the within-hemisphere trials suggesting saccades enhance intrahemispheric processing but not interhemispheric interaction.”

How about

Increased match detection accuracy was found on the within-hemisphere trials as a function of pre-task eye-movements suggesting saccades enhance intrahemispheric processing but not interhemispheric interaction.”

Line 128: After “brain hemispheres” there should not be a full stop before the Lyle & Martin reference.


Line 628: There is a full stop after “ . involve words” that is not needed.

Experimental design

Just a couple of points:

Working Memory Section: From which serial position were the positive probes selected? Was this also just random?

Recognition Memory Section: Were any of the “old” stimuli previously used as positive probes?

Validity of the findings

The discussion of the findings has been revised since the original submission and takes into account the comments of the other reviewers to acknowledge the differences between the present and past experiments (e.g., in terms of methods). This I believe has helped and I have no further comments to make.

Additional comments

As noted above, I think the revised submission is much superior to the original. I feel that my comments have been addressed and the paper has a more coherent feel to it. Some of the ideas outlined and discussion points made would profit from future work and investigation.

·

Basic reporting

Regarding references to the literature, I am uncomfortable with the authors' citation of Christman and Butler's (2005) unpublished conference presentation in support of the claim that "the benefits of SIRE...have been reported to impair the encoding but enhance the retrieval of episodic memories." (Also note that the sentence in which this claim appears is less than eloquent--how can benefits of SIRE impair encoding?) The claim is important and, while I respect the work of Christman and Butler, I feel the claim needs to be backed up with data that a reader can easily access. I would encourage the authors to cite Brunye et al. (2009) instead of, or at least in addition to, Christman and Butler. This will necessitate amending the claim.
Also regarding Brunye et al. (2009), the following description of their findings is unclear: "These...authors found that the eye movements were only beneficial when old-new recognition was compared to forced-choice recognition".

Experimental design

I think it would be helpful for the authors to report in the Subjects section how many subjects scored as left-handed using the handedness inventory.

Validity of the findings

No comment.

Additional comments

When reporting and discussing their findings regarding direction of laterality, I think the authors must acknowledge the small number of left-handers in their sample. For example, the authors state that "when saccades occurred before the working memory task, working memory percent correct accuracy increased significantly as a function of direction of laterality." Could it be that this was driven by the single very low performing left-hander represented by the red triangle in the lower left corner of Figure 3a?
In regard to the aforementioned finding, the authors state that "when saccades were made before encoding scenes in working memory, performance as measured by regression slopes relating overall percent correct with direction of handedness increased significantly positively (Fig 3a) with right-handers (+LQ) performing better than left-handers (-LQ)." This makes it sound like there was a group-level difference between right- and left-handers, and I cannot help but see this as misleading. There were, after all, only two left-handers. Possibly the authors feel their characterization of their findings is defensible, but then I would like to hear their argument.
The authors also state that "recognition performance as measured by both overall percent correct and d-prime in the group of subjects who fixated after working memory and before recognition did increase significantly as a function of direction of handedness, with better performance as LQ increased indicating an advantage for right-handed subjects." Here again the language suggests a group-level difference but there is only one left-handed individual.
In general, I find this version of the manuscript much improved over the original and I applaud the authors' efforts. In my opinion, some revision is still necessary but I think the manuscript is much closer to warranting publication.

·

Basic reporting

.

Experimental design

.

Validity of the findings

.

Additional comments

The revised manuscript satisfactorily addresses the concerns I raised in my initial review. I believe it now merits publication.

---

## Round 0.3 · accepted · Accept

Thank you for your thoughtful responses to reviewers' comments. I think this manuscript makes an interesting contribution to the literature.

#